# Hippocampal place cell sequences differ during correct and error trials in a spatial memory task

Chenguang Zheng [1,2,3✉], Ernie Hwaun[1,4], Carlos A. Loza [1,2] & Laura Lee Colgin [1,2,4✉]

Theta rhythms temporally coordinate sequences of hippocampal place cell ensembles during active behaviors, while sharp wave-ripples coordinate place cell sequences during rest. We investigated whether such coordination of hippocampal place cell sequences is disrupted during error trials in a delayed match-to-place task. As a reward location was learned across trials, place cell sequences developed that represented temporally compressed paths to the reward location during the approach to the reward location. Less compressed paths were represented on error trials as an incorrect stop location was approached. During rest periods of correct but not error trials, place cell sequences developed a bias to replay representations of paths ending at the correct reward location. These results support the hypothesis that coordination of place cell sequences by theta rhythms and sharp wave-ripples develops as a reward location is learned and may be important for the successful performance of a spatial memory task.

[1] Center for Learning and Memory, The University of Texas at Austin, Austin, TX, USA. [2] Department of Neuroscience, The University of Texas at Austin, Austin, TX, USA. [3] School of Precision Instruments and Optoelectronics Engineering, Tianjin University, Tianjin, China. [4] Institute for Neuroscience, The University of Texas at Austin, Austin, TX, USA. ✉email: cgzheng@tju.edu.cn; colgin@mail.clm.utexas.edu

How neurons in the hippocampus, a key brain region for episodic and spatial memory, store and retrieve memories remains incompletely understood. Much evidence supports the theory that rhythmic network patterns in the hippocampus, primarily theta rhythms[1,2] and sharp wave-ripples[3,4], coordinate firing of organized sequences of neurons that represent learned experiences. Theta rhythms coordinate neuronal firing during active exploratory behaviors[5], and sharp wave-ripples coordinate neuronal firing during awake rest and slow-wave sleep[6].

Organized hippocampal neuronal firing sequences that occur during theta rhythms and represent ongoing spatiotemporal experiences are often referred to as "theta sequences"[7–14]. Theta sequences alternate between representing potential future trajectories and current or recent locations[15,16]. In an initial important study, Johnson and Redish[17] showed that CA3 place cell ensembles firing during theta rhythms in a cued-choice task represented potential future paths sweeping ahead of an animal's current location. A subsequent study showed that theta sequences occurring at the start of an animal's journey predict locations extending further ahead of the current location when an animal is traveling to a goal that is further away[12]. Theta sequences have been shown to develop with experience[11], and disruption of theta sequences by medial septum inactivation has been associated with impaired performance of a memory task[14]. These findings are consistent with the hypothesis that theta sequences may be important for correct performance of spatial memory tasks. However, few to no studies have focused on whether theta-organized, temporally compressed sequences of place cells are impaired when animals do not perform spatial memory tasks correctly.

Previous studies have also investigated sequences of place cell firing that occur during sharp wave-ripples (SWRs) in sleep and awake rest[3,18–20]. Such sequences "replay" representations of trajectories from earlier active behaviors on a faster time scale and have been proposed to be important for memory consolidation, memory retrieval, and planning of future trajectories[3,20]. In line with this idea, blockade of neuronal activity during SWRs has been shown to impair memory performance[21–23]. Another study reported that, during periods of immobility in a spatial memory task, place cell ensembles tended to replay trajectories that ended at learned goal locations[24]. A recent study reported that replay of forward-ordered sequences of locations representing animals' future paths toward goal locations was impaired on trials when animals made errors[25]. However, this replay bias was observed during rest periods at a reward well in a spatial working memory task. It is unclear whether similar results would be observed during rest periods prior to errors on a spatial memory task that requires rats to remember spatial locations across a significant delay (i.e., tens of seconds).

In the present study, we set out to investigate whether spatial trajectories represented by temporally compressed sequences of spikes from ensembles of place cells recorded from hippocampal subfield CA1 differed between correct and error trials of a delayed match-to-place task. To assess how representations of trajectories developed across learning, we recorded across multiple trials within a session, keeping the reward location constant within a recording session but changing it across recording sessions. As rats learned reward locations across trials, hippocampal place cell sequences developed that represented paths extending toward the reward location on correct trials. Representations of trajectories extending ahead of the current location were observed on error trials but were less temporally compressed than representations of trajectories on correct trials. During rest periods, place cell sequences developed a significant bias to replay a path that terminated at the correct reward location during correct trials but

not during error trials. Taken together, these results suggest that correct behavioral performance on a spatial memory task is associated with the development of coordinated sequences of place cells during theta rhythms and sharp wave-ripples that represent trajectories extending toward a learned reward location.

## Results

**Behavioral task and performance.** We used a circular track version of a spatial delayed match-to-sample task to investigate whether spatial trajectories coded by organized sequences of place cells differed between the successful and unsuccessful behavioral performance of a spatial memory task (Fig. 1a). Each recording session began with 4–6 "pre-running" trials. In pre-running trials, rats completed laps around the track without receiving rewards. This stage of the task had no memory component but allowed time for experience-dependent changes in hippocampal rhythms[26,27] to reach a stable level and provided a data set that was used to cross-validate a Bayesian decoder (Supplementary Fig. 1). The pre-running trials were also used to verify that rats did not prefer particular locations such as the reward location from the previous day. In the next stage of the task, rats performed 8 trials consisting of sample-test pairs. In the sample-test trials, a reward location on the track was pseudo-randomly selected and remained constant across trials so that learning across trials could be assessed. During the sample phase, the reward location was marked with a visual cue, and the rat was rewarded for stopping at the marked location. In the test phase, the cues were removed, and the rat was required to recall the correct reward location and stop there in order to receive a reward. Sample and test phases were separated by a 30 s delay, and inter-trial intervals were approximately 1 min. Five minutes after sample-test trials were completed, 4–6 "post-test" trials were performed. Post-test trials were identical to the test phase in the sample-test trials (i.e., animals were rewarded for stopping at the same unmarked reward location). Post-test trials were included to test whether rats could recall the reward location after a relatively long delay period. Behavioral performance data from the task showed that rats ($n = 4$) learned the correct reward location within the first few trials and continued to perform significantly above chance during post-test trials (Fig. 1b). Across all recording sessions and rats, the most common errors were stopping one location ahead or behind the correct reward location (Fig. 1c).

No differences were observed in running speeds between correct and error trials (repeated measures ANOVA, main effect of trial types: $F(1,42) = 1.2$, $p = 0.3$, partial $\eta^2 = 0.03$). Moreover, rats were not found to linger for a longer amount of time in the rest box before beginning to run on the test phase of error trials (paired $t$-test: $T(42) = 0.6$, $p = 0.5$, Cohen's $d = 0.1$), nor did they remain at the reward location for a longer amount of time during the sample phase of correct trials (paired $t$-test: $T(42) = 1.5$, $p = 0.1$, Cohen's $d = 0.2$). During the pre-running phase of the task, there was no difference in rats' preferences to spend time at the reward location from the previous day, the current day's reward location, or the current day's erroneous stop locations (repeated measures ANOVA, $F(2,84) = 0.6$, $p = 0.5$, partial $\eta^2 = 0.03$). These results suggest that errors are not explained by differences in rats' behavior or biases for particular locations.

**Place cell sequences representing paths to the reward location developed with experience.** A previous study showed that theta-coordinated sequences of place cells emerged with experience on a novel linear track[11]. We hypothesized that sequences of place cells representing paths toward a reward location in a spatial delayed match-to-sample task would develop as rats learned the reward location across trials. To test this hypothesis, we recorded

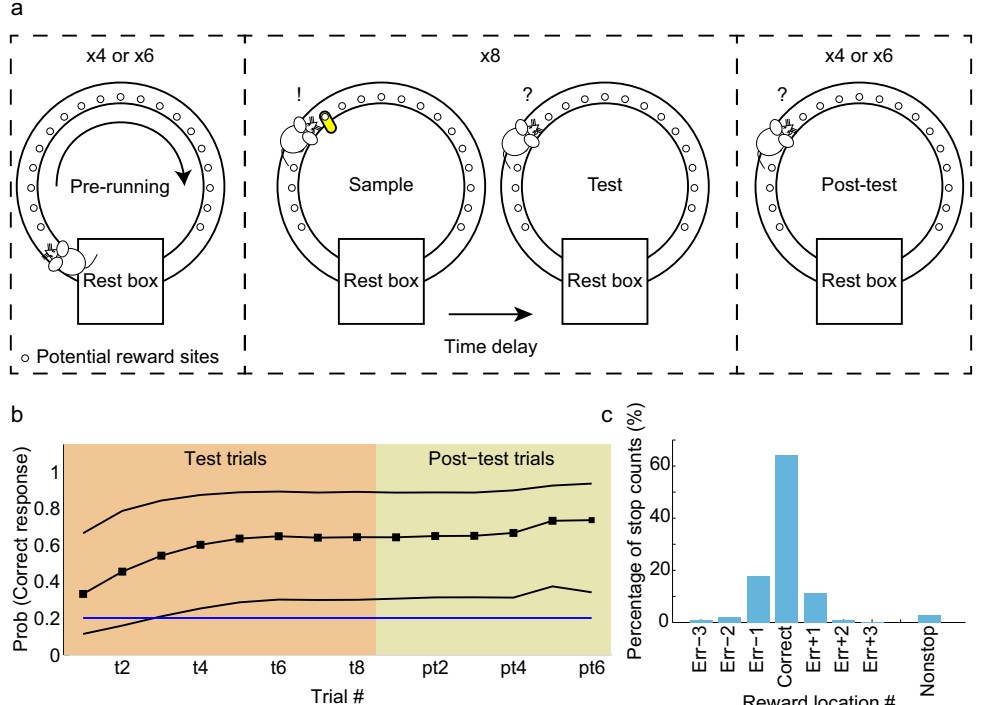

**Fig. 1 Behavioral task and performance. a** Schematic of delayed match-to-sample spatial memory task. Each session consisted of pre-running trials, sample-test trials, and post-test trials. In pre-running trials, rats ran 4 or 6 laps unidirectionally on a circular track without receiving any reward. One of the potential reward sites was pseudo-randomly chosen for each session. In each session, rats performed 8 paired sample-test trials. The correct reward location was marked with a visual cue (yellow marker) during the sample phase. The marker was removed in the test phase, which began 30 sec after the sample phase. After a 5-minute delay, a post-test phase of the task began, which consisted of test trials only (i.e., no cue marked the correct reward location during this phase). Rats received a reward if they stopped at the correct goal location during the sample, test, and post-test trials. The goal/reward location remained constant within a session. **b** Behavioral performance across test (orange, t1-t8) and post-test (beige, pt1-pt6) trials is shown. Black solid lines represent the mean probability of making a correct choice across recording sessions, bounded by 95% confidence intervals. The blue horizontal line indicates chance performance. **c** Proportion of trials in which rats stopped at different locations in test and post-test trials pooled across all recording sessions. Err-n and Err+n ($n = 1, 2, 3$) indicate stop locations that were n locations before or after the correct reward location. Nonstop indicates that rats did not stop at any location in that trial.

ensembles of place cells in hippocampal subregion CA1 of rats performing the delayed match-to-sample task. We then used a Bayesian decoding approach (see Methods; Supplementary Fig. 1) to identify trajectories represented as rats approached the reward location in sample and test phases of sample-test trials. Place cell sequences' representations of paths extending ahead of the animal increased across trials (Fig. 2a; repeated measures ANOVA, main effect of trial number: $F(7,378) = 5.3$, $p = 1.5 \times 10^{-4}$, partial $\eta^2 = 0.4$). This increase was observed for both sample and test phases (correlation coefficient $r = 0.26$, $p = 6.4 \times 10^{-5}$ for sample and $r = 0.19$, $p = 0.005$ for test) and for both correct and error trials (Fig. 2b, c; multiple linear regression, $F(3,436) = 8.1$, $p = 3.1 \times 10^{-5}$, $R^2 = 0.05$; no significant interaction effect between trial type and trial number: $t(436) = -0.5$, $p = 0.6$; no significant effect of trial type: $t(436) = 0.5$, $p = 0.6$; significant effect of trial number: $t(436) = 2.1$, $p = 0.04$). In contrast, place cell sequences' representations of locations extending behind the animal did not increase across trials (Fig. 2d–f; repeated measures ANOVA, significant interaction between location type (i.e., ahead and behind) and trial number: $F(7,756) = 7.1$, $p = 6.3 \times 10^{-7}$, partial $\eta^2 = 0.3$). Sequences representing locations extending behind the animal significantly differed across trials (main effect of trial number for behind locations: $F(7,378) = 2.9$, $p = 0.01$, partial $\eta^2 = 0.3$). It appears as though behind sequences may have gradually decreased across trials; however, successive changes across trials for behind sequences were not statistically significant (multiple linear regression model: $F(3,439) = 2.2$, $p = 0.08$,

$R^2 = 0.02$; no significant trial type by trial number interaction effect: $t(436) = -0.2$, $p = 0.8$; no significant effect of trial type: $t(436) = -0.8$, $p = 0.4$; no significant effect of trial number: $t(436) = -0.3$, $p = 0.8$). Place cell ensemble representations of the overall trajectory from the start location to the stop location remained consistent across trials (Fig. 2g–i; repeated measures ANOVA, no significant main effect of trial number: $F(7,378) = 1.6$, $p = 0.2$, partial $\eta^2 = 0.2$). These results support the conclusion that coordinated sequences of place cells develop representations of paths extending ahead of the animal toward the reward location as they learn the reward location across trials.

**Place cell sequences represented less temporally compressed trajectories as rats approached an incorrect stop location during error trials.** It has been hypothesized that organized sequences of place cells firing within theta cycles are important for retrieving previously stored representations of upcoming trajectories[8,10,28] or encoding of current trajectories[7,8,10,29]. Recent results showed that hippocampal place cell sequences during theta rhythms preferentially represented locations extending ahead of an animal as the animal ran toward a reward location on a spatial alternation working memory task[30]. In the present study, we hypothesized that sequences of place cells during theta-related behaviors would represent paths ahead of the animal differently during correct and error trials in the circle track delayed match-to-sample task. To quantify trajectories

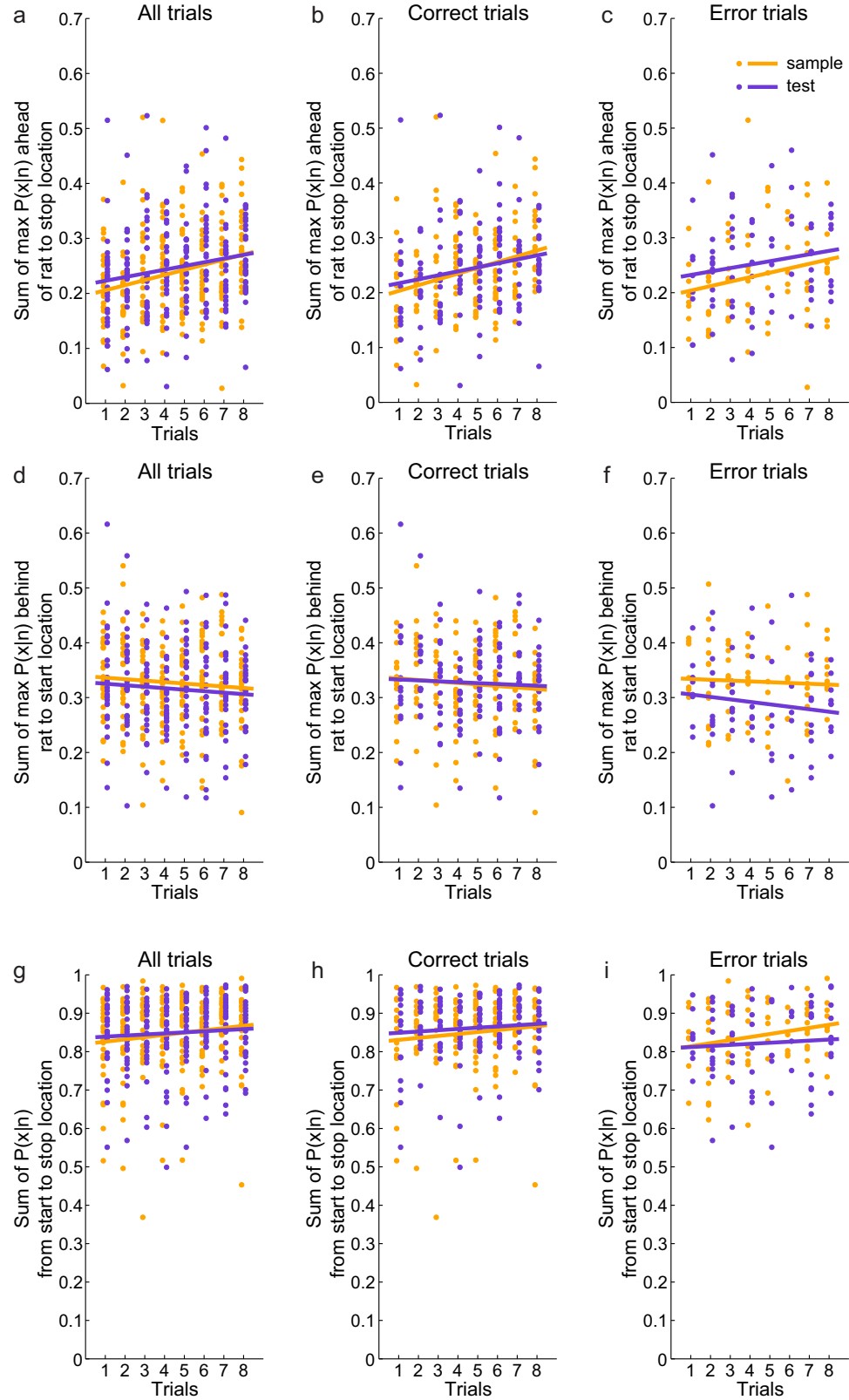

represented during correct and error trials, we decoded place cell ensemble activity as rats ran toward their eventual stop locations. The stop location was the same as the correct reward location on correct and error trials during the sample phase of the task. The stop location was the same as the correct reward location during the test phase of correct trials but corresponded to animals'

incorrect stop locations during the test phase of error trials. We detected epochs that showed significant sequential structure and measured these sequences' slopes as an estimate of sequence compression and direction of decoded paths (see Methods). We hypothesized that place cell sequences representing paths toward the upcoming stop location would be more prominent during the

**Fig. 2 Predictive firing in place cell ensembles developed with learning.** A Bayesian decoder (see Methods) was applied to ensemble spiking activity to estimate posterior probability distributions across sample-test trials. **a–c** The sum of posterior probability ($P(x\,|\,n)$) extending ahead of rats to the stop location (i.e., the sum of the maximum 5 posterior probabilities from 2 bins ahead of the rat's current location to the stop location) as rats ran toward the stop location is shown for each trial (Multiple linear regression: $n = 224$ trials, no significant interaction effect between trial type and trial number: t(436) = −0.5, $p = 0.6$; no significant effect of trial type: t(436) = 0.5, $p = 0.6$; significant effect of trial number: t(436) = 2.1, $p = 0.04$). Regression lines are shown for sample and test trials separately (correlation coefficient: $r = 0.26$, $p = 6.4 \times 10^{-5}$ for sample and $r = 0.19$, $p = 0.005$ for test). Each dot indicates the sum for a lap of the indicated trial type. The sum is shown for all trials (**a**), correct trials (**b**), and error trials (**c**). **d–f** The maximum posterior probability ($P(x\,|\,n)$) estimates behind rats' current positions (i.e., the maximum 5 posterior probabilities from the beginning of the track to a rat's current position) was summed for each lap. In contrast to the results reported in (**a–c**), the sum of posterior probability behind a rat's current position did not increase across trials for all trials (**d**; multiple linear regression model, $n = 224$ trials, F(3,439) = 2.2, $p = 0.08$; no significant trial type by trial number interaction effect: t(436) = −0.2, $p = 0.8$; no significant effect of trial type: t(436) = −0.8, $p = 0.4$; no significant effect of trial number: t(436) = −0.3, $p = 0.8$), correct trials (**e**), and error trials (**f**). **g–i** The sum of the posterior probability ($P(x\,|\,n)$) estimates from the beginning of the trajectory to the stop location is also shown for each lap for all trials (**g**; repeated measures ANOVA: $n = 224$ trials, no significant main effect of trial number: F(7,378) = 1.6, $p = 0.2$), correct trials (**h**) and errors trials (**i**).

test phase of correct trials, considering that place cell sequences developed a representation of upcoming paths as the reward location was learned across trials. In line with this hypothesis, we found that slopes of sequences as rats approached their stop locations were greater for correct test trials than for error test trials (Fig. 3, Supplementary Fig. 2; ANOVA, main effect of trial type (i.e., correct vs. error): F(1,2826) = 15.0, $p = 1.1 \times 10^{-4}$, partial $\eta^2 = 0.005$). These differences in sequence slopes between correct and error trials were not explained by differences in running speeds because running speeds during the approach to the stop location did not differ between correct and error trials (Supplementary Fig. 3). Also, slopes of sequences for different types of error trials (i.e., undershoot errors vs. overshoot errors) were not significantly different (linear mixed model, no significant main effect of error type: F(1,37) = 0.2, $p = 0.7$). This pattern of results suggests that sequence representations of spatial trajectories were more compressed during correct trials than error trials. A separate analysis of trajectory length (x-span) and temporal duration (t-span) of sequences yielded results that were consistent with greater temporal compression of trajectories represented by place cell sequences during the test phase of correct trials (Supplementary Fig. 4a–c). Trajectories represented by sequences during the test phase were longer during correct trials than error trials (Supplementary Fig. 4a, b; ANOVA, main effect of trial type (i.e., correct vs. error) on x-span: F(1,2826) = 13.9, $p = 2.0 \times 10^{-4}$, partial $\eta^2 = 0.005$; main effect of trial type on relative x-span: F(1,2826) = 10.9, $p = 9.5 \times 10^{-4}$, partial $\eta^2 = 0.004$), but the temporal duration of sequences during the test phase did not differ between correct and error trials (Supplementary Fig. 4c; ANOVA, main effect of trial type: F(1,2826) = 1.3, $p = 0.2$, partial $\eta^2 = 4.7 \times 10^{-4}$).

The distribution of error types on the circular track delayed match-to-place task was somewhat skewed toward undershoot errors (Fig. 1c), meaning that trajectories to the stop location during the test phase were often shorter for error trials than correct trials. Thus, it is possible that longer trajectories represented by place cell sequences during the test phase of correct trials compared to error trials reflected greater "look-ahead" distances as rats were heading toward more distant stop locations. This type of effect was shown in a previous study[12], in which place cell sequences represented greater look-ahead distances for more distant goals as rats started their trajectory but not as they approached their stop location. It is possible that a similar effect influenced our results from the test phase because trajectories during the test phase were of different lengths on correct and error trials due to the nature of the task. However, during the sample phase of the task, trajectories to the stop location were identical in length on correct and error trials because the rat stopped at a marked reward location on all trials.

Sample phases in trials after the initial trial likely involve not only encoding of the marked reward location but also recall of the rewarded location from previous trials. Therefore, it is possible that differences in the temporal compression of sequences between correct and error trials would also be present during the sample phase. To test this possibility, we compared slopes of sequences during the sample phase of correct and error trials. As was observed during the test phase, slopes of sequences as animals approached stop locations during the sample phase were different between correct and error trials, with greater slopes again observed during correct trials (Fig. 4; main effect of trial type: F(1,2838) = 15.9, $p = 6.9 \times 10^{-5}$, partial $\eta^2 = 0.006$). Also, similarly to the test phase, the length of trajectories represented by place cell sequences during the sample phase was greater for correct trials than error trials (Supplementary Fig. 4d, e; ANOVA, main effect of trial type on x-span: F(1,2838) = 12.4, $p = 4.3 \times 10^{-4}$, partial $\eta^2 = 0.004$; main effect of trial type on relative x-span: F(1,2838) = 11.6, $p = 6.8 \times 10^{-4}$, partial $\eta^2 = 0.004$). As was observed for the test phase, the temporal duration of sequences during the sample phase did not differ between correct and error trials (Supplementary Fig. 4f; ANOVA, main effect of trial type: F(1,2838) = 0.9, $p = 0.4$, partial $\eta^2 = 3.0 \times 10^{-4}$).

The above-described results suggest that place cell sequences are present during both correct and error trials of a delayed match-to-sample task. However, the results suggest that sequence compression is greater during correct trials than during error trials. This raises the question of what mechanisms may underlie the greater compression of sequences during correct trials compared to error trials. One possibility is that place cells within a sequence do not activate as readily on error trials, perhaps reflecting lower excitatory drive. If so, we hypothesized that place cells' excitation would less readily overcome theta-mediated inhibition[31], and place cell firing at the start of a sequence would commence at a later theta phase. To test this possibility, we assessed the theta phases at which place cell sequences' spikes occurred as rats approached their stop locations during correct and error trials (Fig. 5). Theta phase coding of position was apparent across all trial types. That is, place cells that had place fields at relatively early and relatively late locations fired at relatively early and relatively late theta phases, respectively (see "Categorization of place cell locations" subsection of Methods). Yet, the theta phase distributions differed significantly for correct and error trials (2-way ANOVA Harrison-Kanji test for population means of circular data, the effect of trial type x cell category interaction: $\chi^2(6) = 21.8$, $p = 0.001$). Place cells that represented relatively early locations in the trajectory were more likely to fire at an earlier phase of the theta cycle during correct trials than during error trials (2-way ANOVA Harrison-Kanji test for population means, the significant main effect of trial type (i.e.,

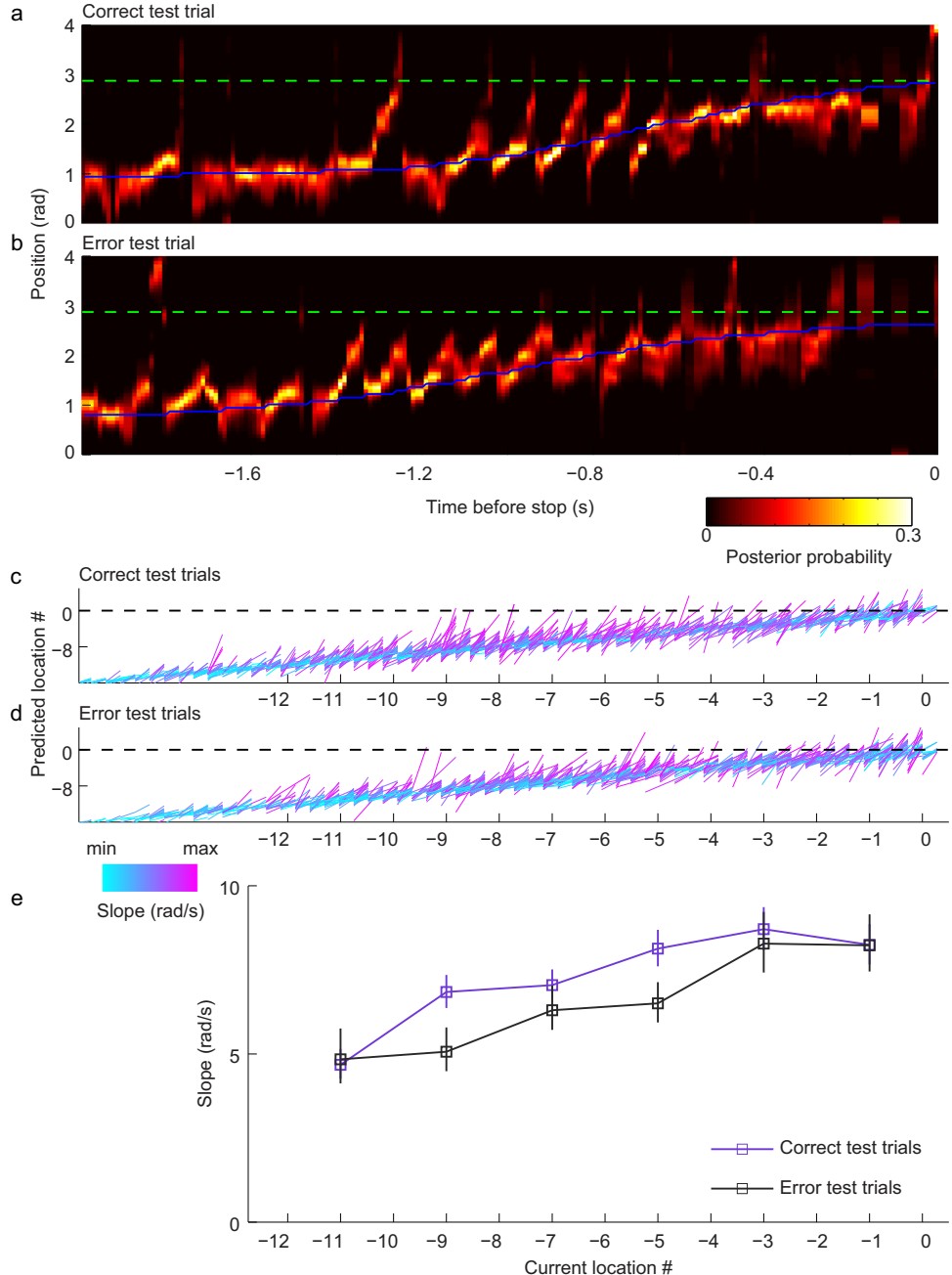

**Fig. 3 Place cell sequences exhibited steeper slopes in correct test trials than in error test trials. a** Example color-coded posterior probability distribution across time from a correct test trial. The green dashed line marks the correct goal location. The blue solid line represents the rat's actual position at each time point. Trajectories were aligned according to the time when the animal stopped (time = 0). **b** Same as (**a**) but for an example error test trial in the same session. The rat stopped one location (~0.2 rad) earlier than the correct goal location in this trial (see Supplementary Fig. 2d for an example trial in which the rat stopped one location too late). **c**, **d** For purposes of illustration and comparison, place cell sequences in correct test trials (**c**) were randomly down-sampled to match the number of place cell sequences in error test trials (**d**). Fitted lines of detected place cell sequences in correct (**c**) and error (**d**) test trials are shown across location numbers as rats approached their stop location (indicated as location 0; same location as correct goal location for correct trials). The x and y coordinates of each sequence's fitted line indicate current position and predicted position (i.e., maximum posterior probability), respectively. The slope of the fitted line for each sequence is shown color-coded. The black dashed line signifies stop location. Decoded locations within a sequence that extended beyond the stop location were included as part of that sequence, and locations in the sequence that extended beyond the stop location were included in the slope calculation. **e** Mean slopes of the lines that were fit to detected place cell sequences are shown across current locations as rats approached their stop location (ANOVA, main effect of trial type (i.e., correct vs. error): $F(1,2826) = 15.0$, $p = 1.1 \times 10^{-4}$, $n = 1879$ positive sequences detected from correct test trials and $n = 959$ positive sequences detected from error test trials). Slope measurements are plotted for the center of each location bin (i.e., locations −12 to −10 plotted at location −11, locations −10 to −8 plotted at location −9, etc.). Data are presented as mean ± error bars. Error bars indicate 95% bootstrapped confidence intervals.

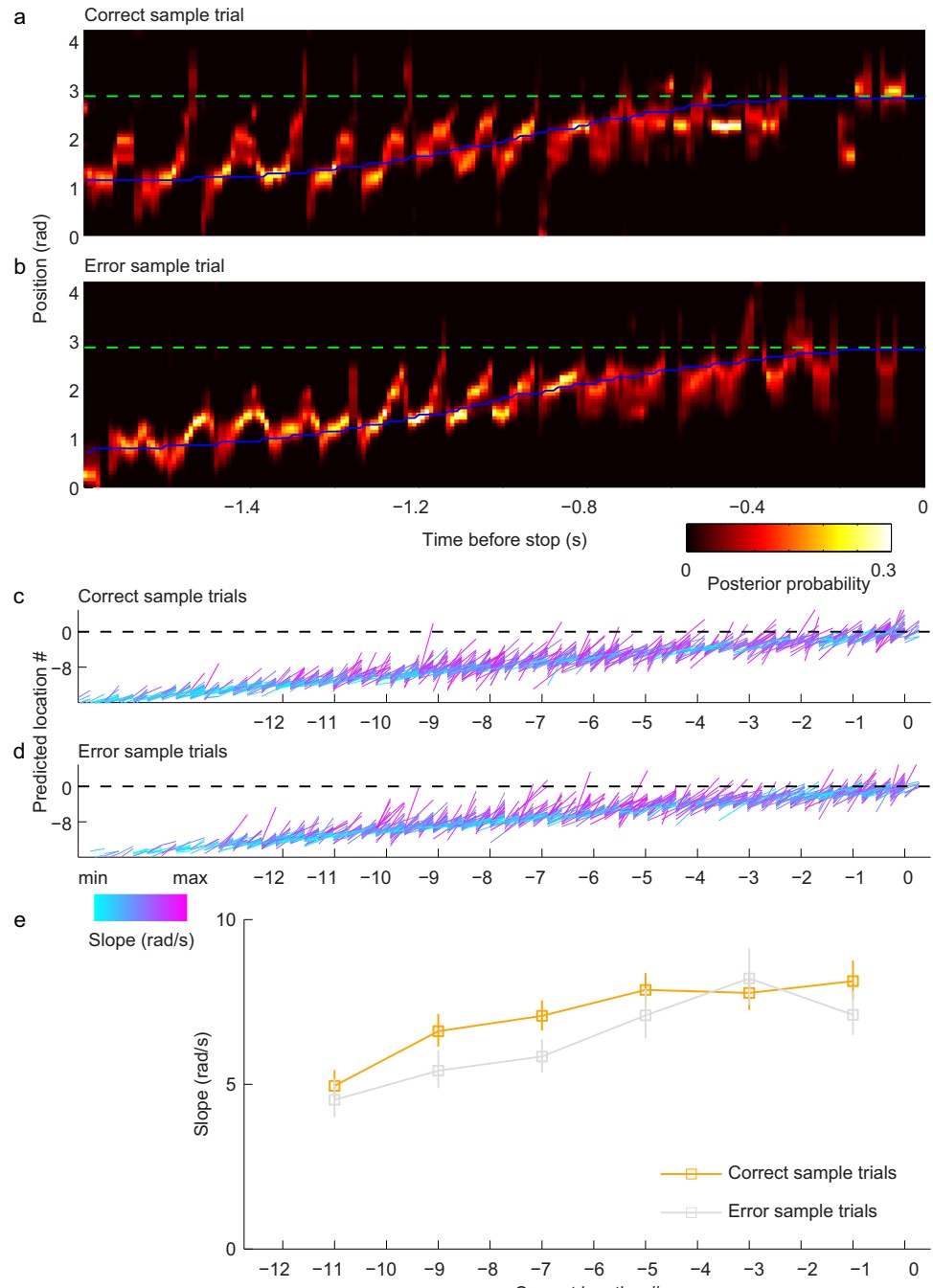

**Fig. 4 Place cell sequences exhibited steeper slopes in correct sample trials than in error sample trials. a–e** Same as (**a–e**) in Fig. 3 but for sample trials. Note that correct and error trials were defined by rats' responses in subsequent test trials (main effect of trial type: F(1,2838) = 15.9, $p = 6.9 \times 10^{-5}$, $n = 1818$ positive sequences detected from correct sample trials and $n = 1032$ positive sequences detected from error sample trials). Data are presented as mean ± error bars (95% bootstrapped confidence intervals) in (**e**).

correct vs. error) for cells that fired early in the sequence: $\chi^2(2) = 7.8$, $p = 0.02$). These results suggest that place cells that start a sequence begin firing at a significantly earlier theta phase during correct trials than during error trials. In contrast, error trials in the spatial delayed match-to-sample task were not associated with abnormal temporal coordination of spikes by slow or fast gamma rhythms (Supplementary Figs. 5 and 6).

For our main analyses, we defined sequences occurring during theta rhythms based solely on ensemble spiking (i.e., epochs in which spikes from place cell ensembles showed significant sequential structure). This allowed us to perform sequence

analyses without making assumptions about the theta phase at which sequences began and ended or about whether sequences remained within a single theta cycle. We also analyzed sequences extending ahead of the animal that were detected within individual theta cycles. Slopes of ahead sequences detected within individual theta cycles were again significantly greater for correct trials than error trials and were not differentially affected by task phase (main effect of trial type: $F(1, 2193) = 7.9$, $p = 0.005$, partial $\eta^2 = 0.004$; no significant interaction between trial type (i.e., correct vs. error) and task phase (i.e., sample vs. test): $F(1,2193) = 1.6$, $p = 0.2$, partial $\eta^2 = 0.001$).

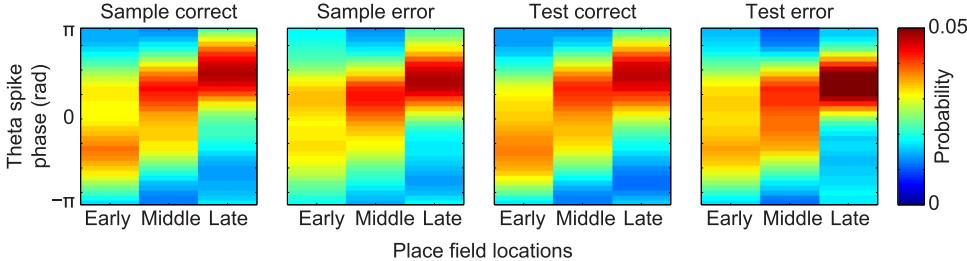

**Fig. 5 Place cells active at the beginning of the trajectory toward the reward fired at earlier theta phases in correct trials than in error trials.** The probability of observing a spike at different theta phases for spikes from place cells with fields located at early, middle, or late locations in the trajectory is shown for each trial type (2-way ANOVA Harrison–Kanji test for population means of circular data, effect of trial type x cell category interaction: $\chi^2$ (6) = 21.8, p = 0.001, n = 19428 spikes; for comparison of theta phases of "Early" cells between correct and error trials, 2-way ANOVA Harrison-Kanji test for population means, significant main effect of trial type (i.e., correct vs. error) for cells that fired early in the sequence: $\chi^2(2) = 7.8$, p = 0.02, n = 8044 spikes). Phase estimates were only obtained from spikes within the place cell sequences detected from 10 to 5 locations before the goal.

For the results described above, only sequences that represented locations extending ahead of the animal (i.e., sequences with positive slopes) were included. Similar results were obtained when sequences with positive slopes and sequences with negative slopes (i.e., sequences that do not represent locations extending ahead of the animal) were combined together (no significant trial type x trial phase interaction: F(1,8833) = 0.2, p = 0.7, partial $\eta^2 = 2.2 \times 10^{-5}$; significant main effect of trial type (i.e., correct vs. error): F(1,8833) = 52.9, p = $3.8 \times 10^{-13}$, partial $\eta^2 = 0.006$ for all sequences combined; significant main effects of trial type (i.e., correct vs. error): F(1,4384) = 28.3, p = $1.1 \times 10^{-7}$, partial $\eta^2 = 0.006$ for test phase and F(1,4449) = 24.5, p = $7.7 \times 10^{-7}$, partial $\eta^2 = 0.005$ for sample phase). Significant differences between correct and error trials were also observed when sequences with negative slopes were analyzed separately (Supplementary Fig. 7; ANOVA, significant main effect of trial type for sample and test phases combined: F(1,3145) = 20.0, p = $8.1 \times 10^{-6}$, partial $\eta^2 = 0.006$; no significant trial type x trial phase interaction: F(1,3145) = 0.4, p = 0.5, partial $\eta^2 = 1.4 \times 10^{-4}$; significant main effects of trial type: F(1,1599) = 7.3, p = 0.007, partial $\eta^2 = 0.005$ for sample phase and F(1,1546) = 13.2, p = $2.9 \times 10^{-4}$, partial $\eta^2 = 0.008$ for test phase). The proportion of positive slope to negative slope sequences was significantly higher when sequences were detected based solely on sequential structure in ensemble spike trains than when sequences were detected within individual theta cycles (5688 and 3169 positive and negative slope sequences, respectively, detected from continuous ensemble spiking; 2565 and 2250 positive and negative slope sequences, respectively, detected within individual theta cycles; $\chi^2(1) = 156.3$, p = $7.3 \times 10^{-36}$).

Surprisingly, differences between slopes in correct and error trials were not observed during the post-test phase. When sequences were detected based on sequential structure, no differences between slopes in correct and error trials were observed during the post-test phase (F(1,1945) = 2.5, p = 0.1, partial $\eta^2 = 0.001$ for positive slope sequences; F(1,1030) = 0.5, p = 0.5, partial $\eta^2 = 5.0 \times 10^{-4}$ for negative slope sequences; F(1,2987) = 2.8, p = 0.1, partial $\eta^2 = 0.001$ for positive and negative slope sequences together). Also, slopes of sequences detected within individual theta cycles during the post-test phase did not differ between correct and error trials (F(1,427) = 0.9, p = 0.3, partial $\eta^2 = 0.002$ for positive slope sequences; F(1,346) = 0.1, p = 0.7, partial $\eta^2 = 3.5 \times 10^{-4}$ for negative slope sequences; F(1,785) = 0.8, p = 0.4, partial $\eta^2 = 0.001$ for positive and negative slope sequences together).

**Replay during awake rest developed a bias to terminate at the reward location during correct but not error trials.** The above

results suggest that organized sequences of place cells during theta rhythms develop as a reward location is learned across trials in a spatial delayed match-to-sample task. The results also suggest that sequences represent spatial trajectories in a more temporally compressed manner during correct trials than during error trials. Place cell sequences that occur during theta-related behaviors are replayed during SWRs[18,32,33]. Replay during SWRs is thought to play a key role in learning and memory, and blocking neuronal activity during SWRs has been shown to decrease performance on memory tasks[21-23]. Thus, we hypothesized that sequence replay during awake SWRs in the rest periods of the circular track delayed match-to-place task would differ between trial outcomes. To assess replay of trajectories, we employed a Bayesian approach to decode place cell ensemble spiking activity recorded when rats were resting (see Methods). We first hypothesized that the fidelity of replay sequences would be poorer during error trials compared to correct trials. In contrast to this hypothesis, we found that place cells replayed trajectories on the circular track with high fidelity during rest periods of both correct and error trials (Fig. 6) for both forward and reverse replay events (Supplementary Fig. 8).

A previous study reported that replay of forward paths toward a goal location in a working memory task was reduced in error trials compared to correct trials[25]. Thus, we hypothesized that trajectories represented during replay would differ between correct and error trials in a delayed match-to-place task. We examined the content of replay sequences during rest periods of sample/test trials and found differences between correct and error trials. Specifically, a bias in replay to terminate at the reward location developed across trials on correct but not error trials (Fig. 7, Supplementary Fig. 9). This bias was observed when rest periods before and after the test phase of trials were grouped together (Fig. 7) and when rest periods before and after the test phase were analyzed separately (Supplementary Fig. 9). There was no significant bias for replay to terminate at the reward location during rest periods in post-test trials (Supplementary Fig. 10a, b). The loss of a replay termination bias could reflect a lack of long-term hippocampal storage of the memory of the reward location. Or, perhaps, enhanced replay of the reward location during the post-test period may be unnecessary, given the relative simplicity of the task. Post-test trials occurred after a 5-min delay, and rats did not show a preference for the reward location in the next recording session (see "Behavioral task and performance" section above). Also, there was no bias in replay observed during the pre-run period prior to learning of a reward location (Supplementary Fig. 10c, d). No significant bias for replay events to start at the reward location was found (Supplementary Fig. 11). Also, no relationship between the representation of a particular end

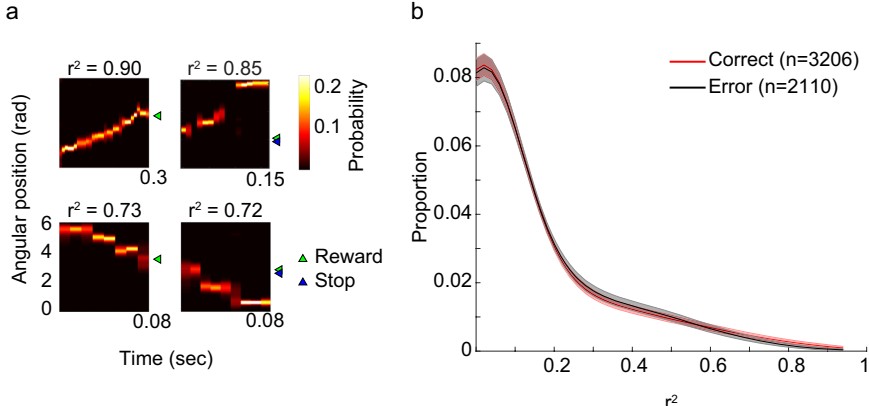

**Fig. 6 Replay quality in the rest box was similar between correct and error trials. a** Example replay events are shown for correct and error trials. The color scale indicates posterior probability from a Bayesian decoder (see Methods), and each example event's $r^2$ value is provided above its probability distribution. Replay events were detected while rats were in the rest box between sample and test trials. The green and blue triangles mark reward and stop positions, respectively, from each trial. **b** Solid lines indicate distributions of $r^2$ values for all detected SWR events in correct and error trials. Shaded error bars indicate 95% bootstrapped confidence intervals.

location during replay events in error trials and animals' subsequent stop locations was found (Fig. 8).

Taken together, these findings suggest that errors in a spatial delayed match-to-sample memory task were associated with abnormalities in place cell sequences. During theta rhythms, sequences represented less temporally compressed spatial trajectories during error trials than during correct trials. Sequences during SWRs in error trials represented trajectories that were less likely to terminate at a learned reward location than on correct trials.

## Discussion

In a spatial delayed match-to-sample task in which a different reward location was learned each day, organized sequences of place cells represented more highly compressed paths as rats moved toward their stop location on correct trials compared to error trials. Also, place cell sequences representing temporally compressed paths extending ahead of an animal toward a goal location developed with learning. Related to this latter result, prior studies have shown that theta sequences are not initially present in a novel environment but develop over time as the environment becomes familiar[11]. Prior work has also shown that place cells' order in theta sequences reverses when animals run backward[34]. Other previous studies have shown that place cell ensembles' representations of upcoming trajectories alternate between possible trajectories as animals approach a choice point or deliberate at a choice point[15,17,30] and represent chosen future trajectories after the choice point[30]. Taken together, these prior results and our present results suggest that organized sequences of place cells emerge with experience and reflect ongoing behaviors. It is possible that transmission of predictive sequences from the hippocampus to downstream structures (e.g., prefrontal cortex and nucleus accumbens) during active theta-related behaviors allows hippocampal memory operations to guide ongoing behavior. In line with this possibility, correlated activity between pairs of prefrontal neurons increases during periods when theta coherence between hippocampus and prefrontal cortex is high[35], and spatial appetitive memory requires projections from CA1 to nucleus accumbens[36]. Also, while hippocampal sequence representations alternate between possible paths before a choice point, prefrontal cortex ensembles consistently represent animals' actual behavior before and after the choice point[30].

The current findings suggest new hypotheses regarding mechanisms underlying errors in spatial memory tasks. On error trials of the spatial delayed match-to-sample task, place cells that coded locations early in a trajectory fired at significantly later phases of ongoing theta cycles compared to correct trials (Fig. 5). These results raise the possibility that delayed firing of place cells that triggered read-out of a sequence was related to incorrect performance of the spatial memory task. Such delayed firing may only allow part of a sequence to be read out before a strong discharge of hippocampal interneurons occurs[1] and terminates the sequence. Or, delayed firing and less temporally compressed sequences could reflect insufficient excitatory drive to the network during theta-related behaviors. In line with this latter possibility, differences in temporal compression of place cell sequences during correct and error trials were also observed for sequences with negative slopes (Supplementary Fig. 7). Insufficient excitatory drive could result from reduced neuromodulatory (e.g., cholinergic) inputs when insufficient attention is paid to the task. Or, insufficient excitatory drive could perhaps result from a lack of synaptic potentiation in connections between cells in a sequence.

Sequence impairments were also observed during rest periods of error trials. Specifically, as rats rested between runs, place cell sequences developed a bias across trials to replay trajectories that terminated at a learned reward location during correct trials but not error trials. A previous study involving rats performing a spatial alternation task showed that place cell co-activation during SWRs was greater during rests preceding correct trials compared to incorrect trials[37]. A subsequent study showed that place cell ensembles during rest replayed trajectories that resembled those that an animal subsequently traversed to a goal location in an open field spatial memory task[24]. A recent study reported that forward replay of place cell sequences representing animals' future trajectories toward a goal location in a spatial working memory task developed with learning and was impaired when animals made errors[25]. Taken together, these findings suggest that replay of place cell ensemble representations of learned trajectories may be important for successful spatial memory performance. This hypothesis is consistent with results showing that disruption of awake SWRs as animals performed a spatial memory task impaired behavioral performance[23]. A recent study employing real-time decoding and disruption of reactivation of place cells representing specific spatial environments supports the hypothesis that replay of particular learned trajectories is required for subsequent recall of memories of those trajectories[38].

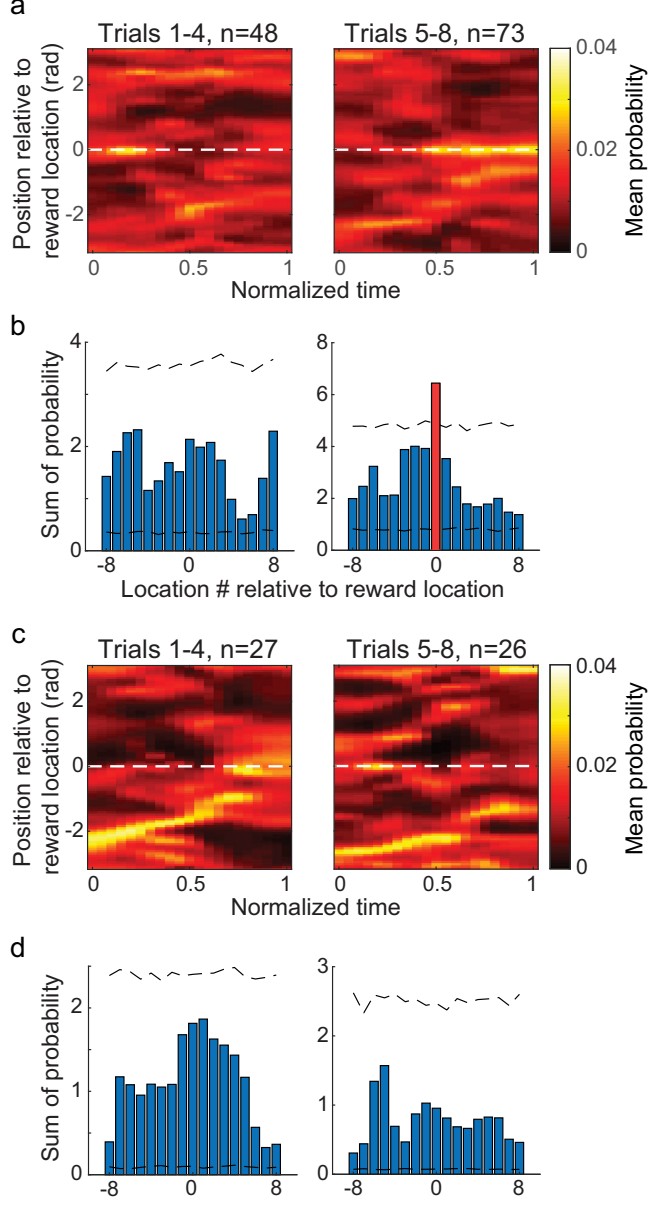

**Fig. 7 A bias for replay events to terminate at the correct goal location emerged after learning during correct trials. a** Mean posterior probabilities of replay events in correct trials. Replay events were aligned to the correct goal location on the vertical axis (i.e., location 0, white dashed line) and normalized time of replay event on the horizontal axis. **b** The sum of posterior probability for the last normalized time bin across replay events for each location number relative to the goal. Dashed black lines mark 95% confidence intervals of a null distribution generated by randomly shifting positions of each replay event (see Methods). A red bar marks the location number with a posterior probability sum that exceeded the corresponding 95% confidence interval (i.e., location 0, which corresponds to the correct goal location. **c**, **d** Same as (**a**, **b**) but for error trials. Note that no positions during error trials showed a posterior probability sum that was greater than the corresponding confidence interval.

The present results do not address the question of whether experience-dependent formation of place cell sequences that represent temporally compressed trajectories during active performance of a spatial memory task precedes the development of replay of the same sequences. However, prior studies shed light

on this question. Although experience-dependent replay during awake rest and coordinated theta sequences emerge at the same developmental stage (i.e., ~P25)[39,40] it is likely that coordinated sequences are initially formed during theta and subsequently activated on a faster time scale during SWRs. Support for this idea was provided by a recent study that recorded place cell ensembles in rats that were passively moved through space[41]. This passive movement served to successively activate place cells while disrupting formation of coordinated theta sequences. Importantly, passive transport and associated suppression of theta sequences blocked replay in subsequent sleep. This result supports the conclusion that linking sequences of place cells during theta is required for subsequent expression of place cell sequences during SWRs. Still, theta sequences can only represent relatively short ongoing trajectories, while SWR-associated replay can include spatially remote and extended trajectories[19,42]. Thus, SWR-associated replay is likely necessary to link multiple sequences together in a temporally compressed manner during the formation of comprehensive spatial and episodic memories.

The differences in hippocampal place cell sequences between correct and error trials discussed above were observed during the sample-test phase of the delayed match-to-sample task. Surprisingly, differences between correct and error trials were not observed during the post-test phase, which occurred after a five-minute delay. Although the reward location changed each day, rats had been trained on the task for many weeks before place cell sequence recordings were obtained. Prior work has suggested that memories that fit within previously stored frameworks ("schema") can be rapidly consolidated into existing memory schema stored in neocortical regions, such as medial prefrontal cortex[43–45]. In this way, memories that are initially hippocampal-dependent can rapidly become hippocampal-independent. It is possible that the memory of the reward location for each day's recording session was rapidly consolidated into neocortical schema during the five-minute rest period following the sample-test phase. Paired recordings from hippocampus and medial prefrontal cortex, and manipulation of activity in these regions, could be used to test this hypothesis but is beyond the scope of the present study.

The current study adds to our understanding of how coordinated sequences of hippocampal place cells relate to performance of spatial memory tasks. Whether similar abnormalities in hippocampal sequences occur in brain disorders associated with memory impairments remains unknown. An interesting hypothesis to test in future studies is whether abnormal place cell sequences contribute to spatial memory impairments in disorders like Alzheimer's disease.

## Methods

**Subjects**. Four male Long-Evans rats weighing from ~400 to 600 g (~4–12 months old) were used in this study. Rats were housed in custom-built acrylic cages (40 cm × 40 cm × 40 cm) on a reverse light cycle (Light: 2000 hours to 0800 hours). The cages contained enrichment materials (e.g., plastic balls, cardboard tubes, and wooden blocks). Active waking behavior recordings were performed during the dark phase of the cycle (i.e., from 0800 hours to 2000 hours). Rats were pre-trained to perform the task prior to recording drive implantation surgery. Rats recovered from surgery for at least one week before behavioral training resumed. During the data collection period, rats were placed on a food-deprivation regimen that maintained them at ~90% of their free-feeding body weight. All experiments were conducted according to the guidelines of the United States National Institutes of Health Guide for the Care and Use of Laboratory Animals under a protocol approved by the University of Texas at Austin Institutional Animal Care and Use Committee.

**Surgery and tetrode positioning**. Recording drives with 20 independently movable tetrodes were surgically implanted above the right hippocampus (anterior-posterior [AP] 3.8 mm, medial-lateral [ML] 3.0 mm, dorsal-ventral [DV] 1 mm on day of implantation). Bone screws were placed in the skull, and the screws and the base of the drive were covered with dental cement to affix the drive to the skull. Two screws in the skull were connected to the recording drive ground. Before

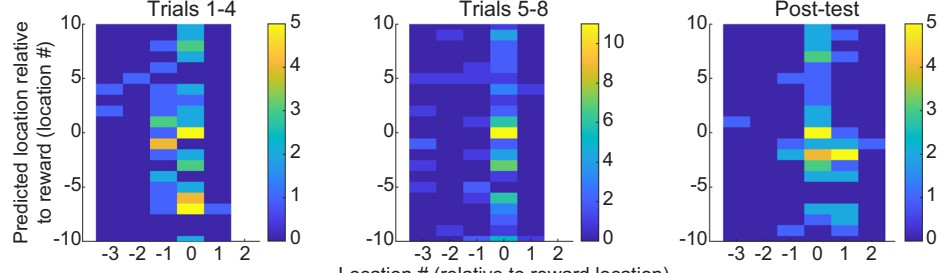

**Fig. 8 Locations represented at the end of replay sequences plotted against animals' actual stop locations.** The y-axis shows the location represented at the end of each replay sequence detected during rest periods of all trials (correct trials correspond to stop location 0 and error trials correspond to other stop locations). The x-axis shows rats' actual stop locations for all trials. Actual stop locations ranging from −3 to −1 indicate that rats stopped at incorrect locations that were 3 to 1 locations before the correct location (i.e., undershoot error trials). Actual locations ranging from 1 to 3 indicate stop locations that were 1 to 3 locations after the correct reward location (i.e., overshoot error trials). Actual locations equal to 0 indicate that rats stopped at the correct reward locations (i.e., correct trials). The colorscale shows the trial counts for each x–y pairing. Results from errors in trials 1–4, trials 5–8, and post-test trials are shown in left, middle, and right panels, respectively. Matthews correlation coefficients were close to zero (equal to 0.0715, 0.0390, 0.0160, respectively, for left, middle, and right panels, with corresponding bootstrap estimates of 0.0713 ± 0.0380, 0.0431 ± 0.0271, 0.0197 ± 0.0349).

surgery, tetrodes were built from 17 μm polyimide-coated platinum-iridium (90/10%) wire (California Fine Wire, Grover Beach, California). The tips of tetrodes designated for single-unit recording were plated with platinum to reduce single-channel impedances to ~150 to 300 kOhms. Tetrodes were gradually lowered to their target locations. Eighteen of the tetrodes were targeted to CA1 stratum pyramidale. One tetrode was designated as a reference for differential recording and remained in a quiet area of the cortex above the hippocampus throughout the experiments. This reference tetrode was moved up and down until a quiet location was found and was continuously recorded against ground to ensure that it remained quiet throughout data collection. Another tetrode was placed in the CA1 apical dendritic layer to monitor and record local field potentials (LFPs) in the hippocampus during placement of the other tetrodes and to later obtain simultaneous recordings from a dendritic layer.

**Data acquisition**. Data were acquired using a Digital Lynx system and Cheetah recording software (Neuralynx, Bozeman, Montana). The recording setup has been described in detail previously[13]. Briefly, LFPs from one channel within each tetrode were continuously recorded in the 0.1–500 Hz band at a 2,000 Hz sampling rate. Input amplitude ranges were adjusted before each recording session to maximize resolution without signal saturation. Input ranges for LFPs generally fell within ± 2,000 to ± 3,000 μV. To detect unit activity, signals from each channel in a tetrode were bandpass filtered from 600 to 6,000 Hz. Spikes were detected when the signal on one or more of the channels exceeded a threshold set at 55 μV. Detected events were acquired with a 32,000 Hz sampling rate for 1 ms. Signals were recorded differentially against a dedicated reference channel (see "Surgery and tetrode positioning" section above). The video was recorded through the Neuralynx system with a resolution of 720 × 480 pixels and a frame rate of 29.97 frames per second. The animal position was tracked via an array of red and blue light-emitting diodes (LEDs) on two of the three HS-27-Mini-LED headstages (Neuralynx, Bozeman, Montana) attached to the recording drive.

**Behavioral task and performance**. Rats were trained to run laps uni-directionally on a circular track (diameter of 100 cm, height of 50 cm, and width of 11 cm) beginning and ending at a wooden rest box (height of 31 cm, length of 28 cm, and width of 21 cm) attached to the track. 19 small white door bumpers were used to mark potential reward locations on the track. Each door bumper was at a distance of 10 cm (or equivalently 0.199 radians) away from its neighboring door bumpers. For every recording session, there were three stages in the behavioral task: Pre-running, Sample-test, and Post-test. In between task stages, rats rested in the wooden box for 5 min. During the pre-running stage, rats ran 4 or 6 laps on the track without receiving any food reward. The Sample-test stage consisted of 8 pairs of sample and test trials. In sample trials, a pseudo-randomly chosen goal location was marked by a yellow plastic cue placed on top of the corresponding door bumper. Of the 19 door bumpers that were on the track, only the 10 door bumpers that were the furthest distance from the wooden box were included as potential reward locations. After a sample trial, rats would rest for 30 s in the wooden box, during which time all cues would be removed by the experimenter (i.e., the yellow plastic cue was removed from the goal location, and potential olfactory cues were removed). Then, rats would run a test trial during which they were required to run toward the unmarked reward location and stop there. One of two tones (2 and 6 kHz sine wave with a duration of 0.25 s) signaling a correct choice or an error would sound if rats stopped at the correct reward location or at a different location. The two tones were randomly assigned to signal a correct or error choice for each rat. Rats received a food reward (i.e., Froot Loops piece) from the experimenter

after the correct tone sounded. Rats had to stop only at the correct goal location to receive a reward. If rats first stopped at a wrong location, no reward would be given in the trial even if they subsequently stopped at the correct goal location. Paired sample-test trials were separated by an approximately 1-minute inter-trial interval and were repeated 8 times. After a five-minute delay, the Post-test phase began. During the Post-test phase, rats ran 4 or 6 more test trials to check retention of their memory for the session's goal location. The goal location remained constant throughout a recording session but was not repeated across successive recording sessions. Rats typically performed one session per day, but 3 out of the 4 rats performed two sessions per day (one in the morning and one in the afternoon) on days when their cell yields were particularly high. A dynamic state-space model[46,47] was implemented to estimate rats' probability of making a correct response during Test and Post-test trials and to compute confidence intervals for the estimated probability. The learning curves were calculated using "Software for Bayesian analysis of learning" (https://health.ucdavis.edu/anesthesiology/research/bayes/index.html), with scripts running in Winbugs (WinBUGS14) in MATLAB. Baseline probability in the model was set to be 0.2 since rats chose one of the 5 door bumpers closest to a goal location in most recording sessions.

**Histology**. Histological sections were prepared in the following manner to verify tetrode positions at the end of experiments. Rats were given a lethal dose of Euthanasia III solution via intraperitoneal injection. Rats then received intra-cardiac perfusion with phosphate-buffered saline followed by formalin to fix brain tissue. Rat brains were then sliced into 30 μm coronal sections and stained with cresyl-violet to confirm final tetrode positions in CA1. In one rat, a poor perfusion precluded localization of each individual tetrode, but damage from the drive implantation was observed in the cortex overlying CA1, and tetrode depths and signals were consistent with placement in CA1.

**Spike sorting and unit selection**. Detected spikes were manually sorted using graphical cluster-cutting software (MClust; A.D. Redish, University of Minnesota, Minneapolis). Spikes were sorted using two-dimensional projections of three different features of spike waveforms (i.e., energies, peaks, and peak-to-valley differences) from four channels. Units with mean firing rates of more than 5 Hz were identified as putative fast-spiking interneurons and were excluded from further analysis. To calculate position tuning for the remaining units, numbers of spikes were divided by time of occupancy at each 0.07-radian position bin on the circular track, followed by smoothing with a Gaussian kernel (standard deviation of 0.14 radians). The running speed for each position ($X_n$) was estimated by taking the difference between the preceding position ($X_{n-1}$) and the following position ($X_{n+1}$) and dividing by the elapsed time (2*1/position sampling frequency). Only times when rats moved at speeds above 5 cm/s were included to estimate position tuning. For all single units, position tunings during and after pre-running were compared (Supplementary Fig. 1a–d). Consistent with previous studies[48,49], the proportion of units active at a goal location increased after learning. Since over-representation of a goal location could potentially bias our decoding analysis (see "Bayesian decoding analysis" section below), units with peak firing rates above 0.5 Hz at a goal location after pre-running were randomly downsampled to match the number of units with peak firing rates above 0.5 Hz elsewhere on the track, excluding the positions that corresponded to the rest box.

**Bayesian decoding analysis**. To translate ensemble spiking activity into angular positions on the circular track, a Bayesian decoding algorithm was implemented[50]. Decoding was performed using a 40-ms sliding time window that shifted 10 ms at

each step. The probability of a rat being at position x, given the number of spikes n from each unit recorded in a time window, was estimated by Bayes' rule:

$$P(x|n) = \frac{P(n|x)*P(x)}{P(n)} \tag{1}$$

where P(n | x) was approximated from the position tuning of each unit after pre-running (i.e., during sample-test and post-test stages). The approximation assumed that the position tuning of individual units were statistically independent and the number of spikes from each unit followed a Poisson distribution[50]. Prior knowledge of position, P(x), was set to 1 to avoid decoding bias to any particular location on the track. The normalizing constant, P(n), was set to ensure P(x | n), or posterior probability, summed to 1. In order to validate the decoding results, the decoder trained from position tunings after pre-running was used to decode ensemble spiking activity during pre-running. Decoding error was defined as the distance between the decoded position with maximum-likelihood and the rat's true position. Cumulative decoding errors for each recording session and their average confusion matrices confirmed the decoding accuracy of the Bayesian decoder (Supplementary Fig. 1e, f). Bayesian decoding analyses were performed using custom routines in MATLAB.

**Detection of place cell sequences**. Place cell sequences during performance of the task were characterized by continuous decoding of ensemble spiking activity that met the following criteria: a sequence included at least 6 consecutive time bins (90 ms) that contained a spike, estimated positions between adjacent time bins did not exceed 1.4 radians, distance between the first and last estimated position within a sequence was more than or equal to 0.07 radians, and there were at least 3 different cells and 5 spikes within a sequence. To compute the slope for place cell sequences, a circular-linear regression line was fitted to the posterior probability distribution[51]. For a sequence to be included in further analyses, at least 60% of the total posterior probability needed to be no more than 0.35 radians away from the fitted trajectory line. In addition, minimal distance between the fitted trajectory and rats' true position had to be less than or equal to 0.35 radians. Sequences were analyzed as rats approached their stop locations. Sequences with positive (Figs. 3 and 4, n = 2838 and 2850, respectively) and negative (Supplementary Fig. 7, n = 1611 for sample phase and n = 1558 for test phase) slopes were analyzed separately. Sequences that began before the stop location but contained decoded locations that extended beyond the stop location were also included. Sequences were not analyzed as animals ran on the track during the "Pre-running" stage because animals had been trained to not stop along the track during this stage (i.e., just to continue running past all potential reward sites).

For almost all analyses throughout the paper, place cell sequences were detected based only on the sequential structure in spike trains, as described above. For the analysis in which sequences were detected within individual theta cycles, the following methods were applied before the sequential structure was detected using the same criteria as described above. Individual theta cycles were cut at the theta phase with the lowest number of spikes (typically at the peak or close to the peak) from all recorded CA1 cells during that session (as in our previous study[13]). Bayesian decoding was performed for those theta cycles containing spikes from at least 3 different cells and at least 5 spikes within the theta cycle. Also, the rat's mean running speed during the time of the theta cycle was required to exceed 5 cm/s (≈0.1 radians/s).

**LFP signal processing**. Theta (6–10 Hz), slow gamma (25–55 Hz), and fast gamma (60–100 Hz) power were estimated from the squared amplitude of the Hilbert transform of their corresponding bandpass-filtered signals. Next, power was normalized across the time when rats were moving (>= 5 cm/s) outside of the rest box, and the normalization was done separately for each frequency band and recording session. Among simultaneously recorded CA1 tetrodes with place cells, the tetrode with the highest raw power in the theta band was selected for normalized power (Supplementary Fig. 6) and phase (Fig. 5 and Supplementary Fig. 5) estimates. Theta, slow gamma, and fast gamma phases were estimated using the angles of their Hilbert transformed signals.

**Categorization of place cell locations**. Positions from the beginning of the track to the reward location were divided into 3 equally spaced segments (i.e., Early, Middle, and Late). Within each place cell sequence, active place cells with peak firing positions inside each position segment were sorted into corresponding categories (Fig. 5). This analysis was designed to test whether the reported difference in slopes between error and correct trials (i.e., Figs. 3 and 4) was explained by sequences during error trials starting at later theta phases or terminating at earlier theta phases than during correct trials. Thus, only spikes from place cell sequences that were active during the period from 10 to 5 positions before the goal location (i.e., the approximate period showing significant differences in slopes between correct and error trials; see Figs. 3e and 4e), were used to obtain the theta phase estimates shown in Fig. 5.

**Detection of SWRs and replay events**. Using LFPs recorded from CA1 tetrodes with place cells, SWRs were detected when rats were in the rest box. The detection method followed that described in a previous study[52]. In brief, LFPs were first

bandpass filtered between 150 and 250 Hz, followed by a Hilbert transform to estimate the instantaneous wave amplitude of the filtered LFPs. After smoothing the amplitude with a Gaussian kernel (standard deviation of 25 ms), a SWR event was detected if the amplitude exceeded 5 standard deviations above the mean for at least 50 ms. Boundaries of detected SWR events were adjusted to first crossings of the mean amplitude. Next, the Bayesian decoder described previously (see "Bayesian decoding analysis" section) was applied to ensemble spiking activity within SWR events. To evaluate how well a posterior probability distribution of positions across time resembled a behavioral trajectory on the circular track, the same circular-linear regression method as described above was used to fit a line to the probability distribution (see "Detection of place cell sequences" section). The corresponding $r^2$ values were used to quantify replay quality following previously published procedures[19,42]. A SWR event was classified as a forward or reverse event if the slope of the fitted line was positive or negative, respectively (Supplementary Fig. 8). A replay event was identified if a SWR event's $r^2$ value was above 0.5, and there were fewer than 20% of time bins without any spikes. Boundaries of a replay event were adjusted to the first and last time bins with spikes inside a SWR event. The adjusted duration of a replay event was required to exceed 50 ms to be included for further analysis. The duration of each replay event was normalized (start of event = time 0 and end of event = time 1). Termination bias of replay events at a goal location was evaluated by aligning different goal locations from each session and summing the last normalized time bin across replay event using a method adapted from a previous study[24] (Fig. 7, Supplementary Figs. 9 and 10). A shuffling method was employed to determine whether termination bias at the goal location was significant. For each replay event, its entire posterior probability across normalized time was circularly shifted 5,000 times by a random distance ranging from 0 to 2π. Summation of the shifted posterior probability across normalized time provided a null distribution to compare against the real data (Fig. 7b, d; Supplementary Figs. 9b, d and 10b, d). The initiation bias of replay events at a goal location (Supplementary Fig. 11) was evaluated using the same method except that the first normalized time bin of the posterior probability was summed across replay events.

Replay events were characterized separately for trials 1–4 and for trials 5–8. Across animals and recording sessions, behavioral performance stabilized at an above chance performance level at trial 5 (Fig. 1b). Therefore, rest periods from trials 1–4 and trials 5–8 were assumed to represent rest periods when learning was incomplete and relatively complete, respectively.

**Estimation of gamma phase shifts across successive gamma cycles**. To measure gamma phase modulation of spike times across gamma cycles, we estimated gamma phases and theta phases of spike times for place cells that spiked within a detected sequence (based on an analysis used by Zheng and colleagues[13]). The gamma cycle with maximal spiking of all simultaneously recorded cells was defined as cycle 0 (Supplementary Fig. 5a, b). Cycles occurring before cycle 0 were numbered with decreasing integer values, and cycles occurring after cycle 0 were numbered with increasing integer values. Incomplete cycles at the beginning or end of a sequence were excluded from analyses. The number of cycles analyzed within each sequence was limited to 3 (cycles −1 to 1) for slow gamma and 5 (cycles −2 to 2) for fast gamma as in a previous study[13]. 2-D histograms of gamma phases and theta phases for spikes from each cycle number were plotted using 30° bins, and were smoothed with a box filter (5 × 5 bins). To determine slow gamma phase shifts across successive slow gamma cycles (Supplementary Fig. 5c), circular-linear correlation coefficients between slow gamma phases and cycle numbers were computed using the 'circ_corrcl' function from the CircStat Toolbox[53] (version 1.21.0.0, https://www.mathworks.com/matlabcentral/fileexchange/10676-circular-statistics-toolbox-directional-statistics). The same method was also used for fast gamma phases (Supplementary Fig. 5f). Phase-locking of spike times to fast gamma was measured by the resultant vector length of fast gamma phases of spikes (Supplementary Fig. 5e), pooled across fast gamma cycles for each trial type. The same method was also used to assess slow gamma phase-locking (Supplementary Fig. 5d).

**Data analysis, statistics, and reproducibility**. Data analyses were performed using custom MATLAB scripts (The Math Works). Paired t-tests, ANOVAs, and repeated measures ANOVAs were performed using standard built-in MATLAB functions (i.e., 'ttest', 'fitlm' and 'fitrm', respectively). Effect sizes are reported as $R^2$ for regression analyses, partial $\eta^2$ for ANOVAs, and Cohen's d (standardized mean difference) for paired t-tests. The Harrison-Kanji test (circular two-factor ANOVA) was performed to test for differences in the theta phase distributions across different trials types and cell categories in Fig. 5 using the 'circ_hktest' function from the CircStat Toolbox[53]. Permutation tests (used for the data shown in Fig. 7b, d and Supplementary Fig. 5c–f) shuffled independent variables (i.e. angular position in Fig. 7b, d and trial type in Supplementary Fig. 5c–f) 5,000 times to generate a null distribution. A result was considered significant if the observed data exceeded the 97.5th percentile or fell below the 2.5th percentile of the null distribution (i.e., a two-tailed test). All confidence intervals were estimated using a standard built-in MATLAB function 'bootci' (n = 5000 resamples). The Matthews correlation coefficient (or phi coefficient) was used to assess whether a correlation was present between rats' stop locations and end locations of replay events (Fig. 8). The Matthews correlation coefficient estimates the correlation between observed and

predicted categorical labels on a scale between −1 and 1. A value of −1 implies total disagreement between prediction and observation, whereas a +1 value represents perfect prediction. Values around 0 imply no correlation. The experimental design of this study did not involve any subject grouping, and thus data collection and analyses were not performed blind. The experiment was performed once; results were not replicated with the second set of rats because of the time and difficulty involved in obtaining large ensembles of place cells in freely behaving rats. Analyses done in this study were verified by two independent experimenters.

**Reporting summary**. Further information on research design is available in the Nature Research Reporting Summary linked to this article.

## Data availability
The datasets used in this study are available at https://web.corral.tacc.utexas.edu/Colgin_Nature_Comms_2021/.

## Code availability
The custom MATLAB scripts used in this study are available at https://github.com/ColginLab/Zheng-et-al-Nat-Comm-2021.

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

## Acknowledgements

We thank A. Akinsooto, S. Dhavala, K. Kallina, and A.J. Wright for recording drive construction, histology, and other outstanding technical support; S. Brizzolara-Dove, G. Kwong, C.G. Orozco, and D. Wehle for help with rat behavioral pre-training; J.B. Trimper for helpful comments on an earlier draft of this manuscript; and all Colgin Lab members for helpful discussions. This work was supported by the National Institutes of Health under award numbers R01 MH102450 (to L.L.C.) and T32 MH106454 (to E.H.), NSF CAREER Award 1453756 (to L.L.C.), and National Natural Science Foundation of China grants 81870847 and 31800889 (to C.Z.).

## Author contributions

C.Z. and L.L.C. designed the experiment; C.Z. and L.L.C. carried out the electro-physiological recordings and behavioral testing; C.Z., E.H., C.A.L., and L.L.C. designed analyses; C.Z. E.H., and C.A.L. wrote analysis programs and analyzed the data; L.L.C., E.H., C.A.L., and C.Z. wrote the paper; and L.L.C. supervised the project. All authors discussed results.

## Competing interests

The authors declare no competing interests.
