## [Peer Review File · Nature Communications]

REVIEWER COMMENTS

Reviewer #1 (Remarks to the Author):

I had the chance to review the present manuscript by Zheng et al. at another journal. The authors have further improved their manuscript, highlighting potential limitations of the task design and the impossibility of comparing correct and error trials of the same length. In my opinion, it is now suitable for publication. I have only one suggestion regarding the analysis of spike-gamma coupling.

The analysis of gamma does not show any differences in terms of phase locking of spikes between correct and error trials. It is a bit strange to end the paper with this. It is worth considering putting these “negative” results in supplementary only as it does not naturally align with the rest of the manuscript, which is primarily focused on cell sequences. One problem is that the authors did only analyze spike-LFP coupling, and it cannot be concluded that gamma has nothing to do with the other effects the authors report. One (out of many others) additional analysis would be to compare theta sequences in theta cycles that show one or the other type of gamma. Even if this analysis turns out negative, it can be included in Figures 3-4 (as an additional panel) while the rest of the analysis (current Fig 8) could be made a supplementary one. The authors can then claim “theta sequences predict reward location, independent of ongoing gamma dynamics” or something similar. If there is an effect, this is even more interesting. If the authors really want to include analyses of gamma in their study, they should also report how low/high gamma evolves with learning, as done with the analysis of position decoding and replay. Otherwise, I don’t think it’s worth talking too much about gamma coupling in the result section. Obviously, the authors should feel free to include this in their discussion.

The authors should consider including Supp Fig 10 as a main figure, it nicely illustrates the effect of replay and learning.

Reviewer #4 (Remarks to the Author):

The manuscript by Zheng et al. analyse the structure of theta and replay sequences during the course of learning a spatial task and perform comparisons between correct and error trials. They find that with learning theta sequences increasingly depict ahead positions towards the goal. This was found for both correct and error trials. Further, the slopes of theta sequences were larger in correct trials than error trials making them more temporally compressed. As a possible mechanism behind this observation the authors suggest that correct trials might be accompanied by a stronger excitatory drive and supported this claim with their analysis on spike theta phases. The authors also analyse replay sequences occurring during rest periods between the trials and find that with learning replay events occurring in correct trials were biased to terminate at the reward location. However, considering the task design, it is unexpected that replay sequences lose this bias in the post-test phase. Nonetheless, in its revised form, this is a carefully analysed study that sheds further light on the debated subject of the role of theta and replay sequences.

I was asked to review the author’s response to the previous reviewer’s comments and add comments of my own if necessary.

Comments:

On page 13 the authors report that when analysing sequences within individual theta cycles a significant difference in sequence slope between correct and error trial was found. However it is not clear whether sequences in this analysis also show larger slope for correct trials since this is not stated in the text or within a figure.

It is unclear whether the theta sequence results from the sample-test phase also hold for the post-test phase, i.e. is there a difference in theta sequence slope between correct and error trials also during the post-test phase?

In the Discussion on page 20 on real time detection and manipulation perhaps the Gridchyn et al., 2020 Neuron study should be cited.

Response to reviewer's comments:

A concern was raised (Reviewer 1) that the degree of how much the animal is "looking ahead" could affect the slope of the theta sequences and that therefore differences in the slope between correct and error trials are a result of differences in how much the animal is looking ahead. The authors were therefore asked to compare correct and error trials of the same length. Due to the nature of the task, correct and error trials within one recording day do not have the same length, prohibiting this type of analysis for the test phases. However, the authors analysed the sequences in the sample phase of correct and error trials (during this phase trials do have the same length) and, although the effect was smaller (3e, 4e), sequences in the sample phase of correct trials had a larger slope. The authors have emphasized this sufficiently in the text and provided a possible explanation for why a difference in sequence slope is already present before the error is made in the test phase.

A concern was raised (Reviewer 1) regarding the author's initial analysis on replay termination position. The new Figure 7 illustrates the maze location where replay events terminate. It now clearly shows that with learning, replay events after correct trials tend to terminate at the goal location.

It was pointed out (Reviewer 1) that replay events during error trials might also be biased to terminate at a specific location. Whilst I have not seen the original figure, the new Supplementary Figure 10 illustrates that replay events after error trials do not terminate in a preferred location.

I agree with Reviewer 1, that it is indeed puzzling that during correct post-test trials replay events do not show a significant bias to terminate at the reward location. One would expect that especially during this phase during which the animal has to remember the reward location without interleaved sample phases, replay ending at the reward location would be enhanced ensuring successful memory recall in subsequent trials. Perhaps repeated replay of the reward location beyond 8 trials is not necessary for performance considering the relative simplicity of the task.

According to Reviewer 2 the previous version of the manuscript did not convey the message of the findings clearly enough. I believe the title gives clear indication of the message of the findings, as does the text.

Reviewer 2 asks about the meaning and behavioural relevance of an increased slope in theta sequences. The authors have explained in the manuscript that the increased slope is an expression of theta sequences being temporally more compressed possibly reflecting increased excitatory drive. The authors also provide explanation for why temporally compressed sequences would be tied to correct task performance.

The questions regarding Figure 2 were resolved and the authors clearly explained the underlying analysis.

The authors have appropriately addressed the comment about the statistical test used for Figure 5 and have now applied a 2-way test for circular data followed by a post-hoc test.

We thank the reviewers for their constructive comments, which we believe have improved our manuscript. We were able to address all comments with text revisions and additional analyses. The submitted manuscript indicates our revisions in red text (minor changes such as reordering of figure numbers, renumbering of references, correcting typos, etc. are not indicated in red text). Please see our responses to specific review comments below. Reviewers' comments are italicized, and our responses are in plain text.

Reviewer #1:

I had the chance to review the present manuscript by Zheng et al. at another journal. The authors have further improved their manuscript, highlighting potential limitations of the task design and the impossibility of comparing correct and error trials of the same length. In my opinion, it is now suitable for publication. I have only one suggestion regarding the analysis of spike-gamma coupling.

We thank the reviewer for their positive comments.

The analysis of gamma does not show any differences in terms of phase locking of spikes between correct and error trials. It is a bit strange to end the paper with this. It is worth considering putting these "negative" results in supplementary only as it does not naturally align with the rest of the manuscript, which is primarily focused on cell sequences. One problem is that the authors did only analyze spike-LFP coupling, and it cannot be concluded that gamma has nothing to do with the other effects the authors report. One (out of many others) additional analysis would be to compare theta sequences in theta cycles that show one or the other type of gamma. Even if this analysis turns out negative, it can be included in Figures 3-4 (as an additional panel) while the rest of the analysis (current Fig 8) could be made a supplementary one. The authors can then claim "theta sequences predict reward location, independent of ongoing gamma dynamics" or something similar. If there is an effect, this is even more interesting. If the authors really want to include analyses of gamma in their study, they should also report how low/high gamma evolves with learning, as done with the analysis of position decoding and replay. Otherwise, I don't think it's worth talking too much about gamma coupling in the result section. Obviously, the authors should feel free to include this in their discussion.

In the revised submission, we have taken the reviewer's suggestion and moved the gamma results presented in Figure 8 to the Supplementary Figures (new Supplementary Figures 5 and 6). We thank the reviewer for this suggestion, which allowed us to comply with the 5000 word limit of Nature Communications (our prior submission was over the limit by >600 words).

The authors should consider including Supp Fig 10 as a main figure, it nicely illustrates the effect of replay and learning.

In the revised submission, we have taken the reviewer's suggestion and moved Supplementary Fig. 10 from the last submission to the main figures in the revised submission (new Figure 8).

Reviewer #4:

The manuscript by Zheng et al. analyse the structure of theta and replay sequences during the course of learning a spatial task and perform comparisons between correct and error trials. They find that with learning theta sequences increasingly depict ahead positions towards the goal. This was found for both correct and error trials. Further, the slopes of theta sequences were larger in correct trials than error trials making them more temporally compressed. As a possible mechanism behind this observation the authors suggest that correct trials might be accompanied by a stronger excitatory drive and supported this claim with their analysis on spike theta phases. The authors also analyse replay sequences occurring during rest periods between the trials and find that with learning replay events occurring in correct trials were biased to terminate at the reward location. However, considering the task design, it is unexpected that replay sequences lose this bias in the post-test phase.

Nonetheless, in its revised form, this is a carefully analysed study that sheds further light on the debated subject of the role of theta and replay sequences.

We thank the reviewer for their positive comments.

I was asked to review the author's response to the previous reviewer's comments and add comments of my own if necessary.

Comments:

On page 13 the authors report that when analysing sequences within individual theta cycles a significant difference in sequence slope between correct and error trial was found. However it is not clear whether sequences in this analysis also show larger slope for correct trials since this is not stated in the text or within a figure.

We thank the Reviewer for catching this oversight. Slopes of sequences within individual theta cycles were greater for correct trials than error trials, and this has now been clarified in the revised submission (line 258).

It is unclear whether the theta sequence results from the sample-test phase also hold for the post-test phase, i.e. is there a difference in theta sequence slope between correct and error trials also during the post-test phase?

Differences in theta sequence slopes between correct and error trials were not observed during the post-test phase. These results have been included in the revised manuscript (lines 283-292), and a potential explanation with supporting citations is now provided in the Discussion (lines 418-432).

In the Discussion on page 20 on real time detection and manipulation perhaps the Gridchyn et al., 2020 Neuron study should be cited.

We apologize that we did not cite this very important and relevant work in our original submission. This paper has now been included in the Discussion of the revised manuscript (lines 395-398).

Response to reviewer's comments:

A concern was raised (Reviewer 1) that the degree of how much the animal is "looking ahead" could affect the slope of the theta sequences and that therefore differences in the slope between correct and error trials are a result of differences in how much the animal is looking ahead. The authors were therefore asked to compare correct and error trials of the same length. Due to the nature of the task, correct and error trials within one recording day do not have the same length, prohibiting this type of analysis for the test phases. However, the authors analysed the sequences in the sample phase of correct and error trials (during this phase trials do have the same length) and, although the effect was smaller (3e, 4e), sequences in the sample phase of correct trials had a larger slope. The authors have emphasized this sufficiently in the text and provided a possible explanation for why a difference in sequence slope is already present before the error is made in the test phase.

We thank the reviewer for their positive comments.

A concern was raised (Reviewer 1) regarding the author's initial analysis on replay termination position. The new Figure 7 illustrates the maze location where replay events terminate. It now clearly shows that with learning, replay events after correct trials tend to terminate at the goal location.

We thank the reviewer for their positive comments.

It was pointed out (Reviewer 1) that replay events during error trials might also be biased to terminate at a specific location. Whilst I have not seen the original figure, the new Supplementary Figure 10 illustrates that replay events after error trials do not terminate in a preferred location.

We thank the reviewer for their positive comments. Please note that Supplementary Figure 10 has been moved to the main text (new Figure 8), based on the recommendation of Reviewer 1.

I agree with Reviewer 1, that it is indeed puzzling that during correct post-test trials replay events do not show a significant bias to terminate at the reward location. One would expect that especially during this phase during which the animal has to remember the reward location without interleaved sample phases, replay ending at the reward location would be enhanced ensuring successful memory recall in subsequent trials. Perhaps repeated replay of the reward location beyond 8 trials is not necessary for performance considering the relative simplicity of the task.

We thank the reviewer for suggesting this alternate interpretation. We have added this interpretation to the Results section of the revised manuscript (lines 324-326).

According to Reviewer 2 the previous version of the manuscript did not convey the message of the findings clearly enough. I believe the title gives clear indication of the message of the findings, as does the text.

We thank the reviewer for their positive comment.

Reviewer 2 asks about the meaning and behavioural relevance of an increased slope in theta sequences. The authors have explained in the manuscript that the increased slope is an expression of theta sequences being temporally more compressed possibly reflecting increased excitatory drive. The authors also provide explanation for why temporally compressed sequences would be tied to correct task performance.

We thank the reviewer for their positive comment.

The questions regarding Figure 2 were resolved and the authors clearly explained the underlying analysis.

We thank the reviewer for their positive comment.

The authors have appropriately addressed the comment about the statistical test used for Figure 5 and have now applied a 2-way test for circular data followed by a post-hoc test.

We thank the reviewer for their positive comment.